# Validation of the Block Walk Method for Assessing Physical Activity occurring on Sidewalks/Streets

**DOI:** 10.3390/ijerph16111927

**Published:** 2019-05-31

**Authors:** Richard R. Suminski, Gregory M. Dominick, Eric Plautz

**Affiliations:** 1Center for Innovative Health Research, Department of Behavioral Health and Nutrition, University of Delaware, Newark, DE, 19726, USA; 2Department of Behavioral Health and Nutrition, University of Delaware, Newark, DE, 19726, USA; gdominic@udel.edu (G.M.D.); eaplautz@udel.edu (E.P.)

**Keywords:** observation method, public health, health behavior, measurement

## Abstract

The block walk method (BWM) is one of the more common approaches for assessing physical activity (PA) performed on sidewalks/streets; however, it is non-technical, labor-intensive, and lacks validation. This study aimed to validate the BWM and examine the potential for using a wearable video device (WVD) to assess PA occurring on sidewalks/streets. Trained observers (one wearing and one not wearing the WVD) walked together and performed the BWM according to a previously developed protocol along routes in low, medium, and high walkable areas. Two experts then reviewed the videos. A total of 1150 (traditional) and 1087 (video review) individuals were observed during 900 min of observation. When larger numbers of individuals were observed, the traditional method overestimated the overall number of people as well as those walking and sitting/standing, while underestimating the number of runners. Valid estimates of PA occurring on sidewalks/streets can be obtained by the traditional BWM in low and medium walkability areas and/or with non-common activities (cycling); however, its validity is questionable when sidewalks/streets use volume is high. The use of WVDs in PA assessment has the potential to establish new levels of accuracy, reduce resource requirements, and open up the possibility for retrospective analysis.

## 1. Introduction

Physical inactivity facilitates the development of chronic diseases including obesity, cardiovascular disease, type 2 diabetes, and some cancers, and independently contributes substantially to the total amount spent on annual US healthcare [1,2,3,4]. Although efforts to increase physical activity (PA) have met with some success, half of US adults are still not sufficiently active to achieve health benefits and a significant proportion of children under age 19 do not engage in recommended amounts of PA [3,4]. Whereas psychosocial factors associated with PA are traditionally targeted within behavioral interventions, such approaches are limited without considering the contextual influence on PA [5]. Indeed, there is now clear evidence that a substantial proportion of PA is determined by aspects of the social and physical environments within community settings [5,6,7,8].

In the past 15 years, there has been an exponential rise in studies looking at social determinants of health, especially those in the neighborhood and built environments that could affect PA [5,9,10]. Neighborhood built environment characteristics such as mixed-land use, access to destinations, population density, and street/sidewalk connectivity are shown to favorably influence pedestrian PA [11,12,13]. Moreover, some built environment characteristics are shown to influence the degree to which individuals engage in PA [14,15,16]. In particular, access to neighborhood sidewalks/streets is associated with greater participation in moderate-to-vigorous PA [17,18,19]. Sidewalks/streets are among the most common aspects of the built environment where a considerable proportion of outdoor, physical activities (e.g., walking, running, cycling) are performed largely within neighborhoods that are proximal to a person’s home [18,20,21]. For example, approximately 70% of adults who engage in recreational walking report using the sidewalks/streets in their neighborhood and adults who are PA near their homes gain about 17% more time in daily moderate-to-vigorous PA [20,21].

Studies and evaluations of physical activities performed on sidewalks/streets, whether to detect changes in usage or determine how associated environmental conditions impact their usage, necessitate a reliable, accurate, and easily administered approach for assessing PA. Self-report questionnaires are hampered by recall bias, plus they have not been adequately validated for geo-locating physical activities [21,22,23]. This is particularly true when asking respondents if they were physically active on the sidewalks/streets in their neighborhood [10,21,24]. Objective measures including accelerometers and pedometers, combined with global positioning systems (GPS), have been used to geo-locate physical activities [25,26]. Although these offer an improvement over self-report questionnaires, drawbacks exist. First, the logistics and cost to use these in community-level evaluations is prohibitive. Second, although accelerometers and pedometers are effective for measuring temporal changes in PA and sedentary behaviors, data from these devices provide no contextual information regarding the type of PA individuals perform, nor the location in which PA occurs. Even when accelerometer data are coupled with GPS, only the sample of individuals (cohort) wearing the monitor/GPS systems are counted, and their PA data is restricted to the geographical locations they visited. As with recall questionnaires, monitor/GPS systems are not useful for determining utilization rates of specific geographical areas such as sidewalks/streets.

In contrast, the observation method is a reliable approach to counting the number of individuals engaged in various physical activities in different environmental settings including parks as well as sidewalks/streets [27,28]. It is also widely employed by transportation departments for the purpose of counting pedestrians. The pedestrian count (PC) involves a stationary observer who records the volume and direction of pedestrian traffic along various routes [29]. An extension of the PC is the block walk method (BWM) [30,31]. The BWM uses a mobile, time sampling technique in which observers walk along pre-defined sidewalk/street segments at a set pace while systematically recording the number of individuals observed engaging in different types of PA (e.g., walking, cycling, etc.). The BWM is an improvement over PC because it provides an objective, scientifically rigorous, and replicable method to assess PA on sidewalks/streets. Unlike the static PC approach, the BWM relies on mobile observations that capture a substantially greater proportion of sidewalks/streets in which contextually rich information on PA and environmental exposures is collected across diverse geographical settings and environmental conditions. Moreover, it is reliable, and physical activities assessed with the BWM are significantly associated with micro-level environmental characteristics (e.g., sidewalk defects, crosswalks) [30,31,32,33]. Despite the BWMs many benefits, it has not been updated since its introduction in 2006 and limitations inherent in its original design are still present. In its current form, the BWM is time consuming, requires extensive training, and has questionable accuracy when observing large groups at one time.

Today’s video technology affords unprecedented opportunities to obtain high-resolution, georeferenced imagery. Wearable video devices (WVD) could feasibly be used to assess sidewalk/street users in real-time, thus providing a means to validate the BWM and eventually augment it with video technologies. Although video capture has not been used to study human physical activity in unconstrained conditions (i.e., on sidewalks/streets), it has been utilized to assess human movement in controlled settings (e.g., laboratory) and examine animal behaviors in the wild (e.g., Orangutan nest construction), glacial change, and traffic patterns [34,35,36,37,38]. Researchers have found the use of videos to be less costly, more efficient, and more precise than traditional observation approaches [36]. The adaptation of current video technology to the study of PA behavior on sidewalks/streets is a logical next step in the evolution of PA measurement. Therefore, the current study sought to utilize WVD technology to examine the validity of the traditional BWM and demonstrate the utility by which WVDs can be used to quantify PA occurring on sidewalks/streets.

## 2. Materials and Methods

**Overview:** This cross-sectional study was conducted in Newark, DE from June to August when temperatures ranged from 63 to 84 °F. The BWM was completed six times along three observation routes representing low, medium, and high walkability. Walking side-by-side, two observers performed the BWM simultaneously with one observer following the traditional BWM procedures and the other recording video using the WVD. Two investigators independently reviewed the videos and reached consensus on the number of individuals being physically active along the observation routes. Comparative analyses were conducted to determine the equivalence of the two approaches. The study was conducted in accordance with the Declaration of Helsinki and the protocol was approved by the Ethics Committee of University of Delaware (Project identification code 1134582-2).

**Walkability:** The WalkScore^®^ search engine was used to measure the walkability of the observation routes where the BWM was conducted [39]. This process entailed selecting an address in an area known by the researchers to possess a certain degree of the qualities used to determine WalkScore. For instance, an address along the main shopping route was selected because the area contained a large number of different types of retail shops, which is used in the WalkScore algorithm. From this address, 1000 ft. segments following the sidewalks/streets were mapped using the ruler tool in Google Earth (a geobrowser that accesses satellite, aerial imagery, and other geographic data to represent the Earth as a three-dimensional globe). The ruler tool is a geographical information systems-based application with sub-meter resolution. The WalkScore of the segment was then determined by obtaining the WalkScore for five randomly selected addresses along the segment. This process was repeated until one continuous, 5000-observation route consisting of five segments with WalkScores falling within the low (WalkScore < 33), medium (WalkScore 33 to <66), and high (WalkScore > 66) walkability ranges.

We used the WalkScore because it is a valid measure for estimating walkability [40,41]. It is significantly correlated with GIS-derived indicators of neighborhood walkability such as the availability of retail destinations, intersection density, amenities, street connectivity, residential density, and access to public transit provisions [40,41]. In addition, a higher Walk Score^®^ is significantly associated with minutes/week of transport and leisure walking independent of socio-demographic and health variables [42]. WalkScore uses publicly available data from various sources (Google, Education.com, Open Street Map, and Localeze) and an algorithm to assign a score to a location based on the straight-line distance to various categories of amenities (e.g., schools, stores, parks, and libraries) weighted by facility type priority and a distance decay function [43]. The result is a walkability score between 0 and 100, with 0 being the least walkable and 100 being the most walkable. The location can be entered as geographic coordinates or as an address, which is then geolocated using Google Geolocation [39].

**Observation Schedule:** Each observation route was observed on three randomly selected days of the week during three observation periods per day. Each observation period lasted 50 min and occurred during each of the following time periods: 08:00–09:00, 12:00–13:00, and 17:00–18:00 (*note: all observations occurred during daylight hours*). Observations were not performed on days having an event that would affect counts (e.g., parade, marathon) or during times when it was raining or snowing.

**BWM Procedures:** During an observation period, two trained observers (one wearing a WVD and the other not wearing a WVD) traversed an observation route at a pace of 100 ft/min (50 steps/min (largo); stride width 2 ft; pace set by a cell phone metronome). The dual observer format was used because the BWM requires an observer to look away from the observation field while entering data on the BWM instrument, and this has been found to be a source of error especially with larger groups [30,31,32]. Our intent was to eliminate this error from the video recordings.

The observer without the WVD recorded the number of individuals engaging in the targeted activities within an observation field. The observation field was defined as a line extending to the left and right of the observer’s shoulders, linear and perpendicular to the observer’s plane of motion. The observation fields ranged in width from 30–70 ft. and included both sidewalks (if present) and the streets associated with the observation routes. Individuals were counted only if they crossed a parallel plane of motion with the observer. For example, individuals walking down the sidewalk towards the observer (from ahead or from behind the observer) were counted if they continued to walk past the observer. The observer made every attempt to count an individual only once during an observation period. When an observer encountered a street intersecting the observation route being observed, they ceased observing, crossed the street, and then resumed observations. An observation recording instrument was previously developed specifically for the procedure [30]. The instrument was designed so that an observer could record the PA observed, the street name where the PA occurred, and the number of individuals engaged in the PA.

**Expert Video Review of Videos:** A two-step process was used to manually analyze the WVD videos obtained during the BWM. First, the reviewers conducted independent evaluations of the videos using the BWM criteria to count individuals walking, cycling, running, and sitting/standing along the observation routes. Next, they convened to review their findings and reach a consensus.

**Observer Training:** A total of four field observers participated in two training sessions prior to beginning data collection. During the first training session, they were given detailed instructions on the BWM and procedures to be used. The second training session involved mock field observations.

**Meteorological Conditions:** Data on meteorological conditions (rainfall, relative humidity, temperature, wind speed, and barometric pressure) for the exact time of day observations were conducted were obtained from an automated weather sensor system (AWSS) located at the local airport.

**Wearable Video Device—Pivothead Smart (Pivothead, Denver, CO, USA):** The Pivothead Smart is one of the most technologically advanced WVDs (Figure 1 and Figure 2). The camera is centered in the bridge of the glasses for the truest first-person perspective possible and it features an 8 MP Sony CMOS sensor for capturing full 1080p HD MP4 video at 30 frames per second as well as 8MP stills (Figure 3). A 32 GB memory card can be used, providing up to 8 hours of video recording per card at 1080p, and they possess a self-contained battery, allowing six to eight hours of recording time. The field of vision is 77 degrees, which approximates the human 90-degree field of vision, and they can be fitted with polarized prescription RX lenses. The Pivothead also allows for audio recordings, time and date stamps, and geolocation capabilities.

### Statistical Analysis

The relationship between walkability category (1 = low, 2 = medium, 3 = high walkability) and total observed was examined using Pearson product moment correlation. Bland–Altman plots and intraclass correlation coefficients (ICCs) were used to assess the agreement between the traditional BWM and expert reviews of BWM video regarding the counts of people walking, running, cycling, and sitting/standing within an observation period. Inter-rater reliabilities, considering the degree of correlation and agreement between measurements, were assessed using ICCs estimated from multiple rater, consistency, 2-way fixed-effects models [44]. Statistical power was 0.8, given 18 observation periods, two observations/observation period, alpha = 0.05, acceptable reliability = 0.7, and expected reliability = 0.9. Intraclass correlations coefficient estimates less than 0.5 were indicative of poor reliability, values between 0.5 and 0.75 indicated moderate reliability, values between 0.75 and 0.9 indicated good reliability, and values greater than 0.90 indicated excellent reliability [45].

For the Bland–Altman analysis, pairwise comparisons were made between the traditional BWM and expert reviews of BWM videos for people walking, running, cycling, and sitting/standing [46]. Means and differences between each method were calculated for each observation period and plotted against each other with the means between methods on the horizontal axis and the differences between methods on the vertical axis. The mean differences, as well as the 95% limits of agreement (mean of the differences (*đ*) ± 1.96 × standard deviation), were also added to the Bland–Altman plots [47]. Linear regression analysis was used to examine the relationship between mean differences and averages of the methods. All statistical analyses were performed using the SPSS statistical software package (IBM Corp. 2015. SPSS Statistics for Windows, Version 23.0. IBM Corp. Armonk, NY, USA) with alpha set a priori at 0.05.

## 3. Results

The BWM was conducted for a total of 900 min (50 min × 18 observation periods; 600 min on weekdays and 300 min on weekend days). Expert reviews of the BWM videos indicated that 0 walkers were counted during one observation period, 0 runners were counted during seven observation periods, 0 cyclists were counted during three observation periods, and 0 sitters/standers were counted during 10 observation periods. For all 18 observation periods, individuals were observed and counted (range 3 to 340 individuals/50 min observation period). Walkability (1 low–3 high) was highly correlated with video review measures of the total observed (r = 0.70; *p* < 0.005), number walking (r = 0.71; *p* < 0.005), and the number sitting/standing (r = 0.76; *p* < 0.001), but not the numbers running (r = 0.29; *p* = 0.24) or cycling (r = 0.27; *p* = 0.28). Nearly identical outcomes were found when traditional counts were correlated with walkability (results not shown).

The counts of individuals engaged in different types of PA as well as the total number of counts from the traditional BWM and expert reviews of BWM videos are summarized in Table 1. Both methods were fairly similar with regards to the total number of individuals engaged in each PA type and overall with the difference in total number observed being 63 individuals (an overestimation by traditional of 5.8%). The traditional method overestimated the total number of people walking and sitting/standing and for overall counts of people observed, while underestimating the number of runners. The number of cyclists counted was exactly the same for both methods. Walking (according to both methods) was by far the most common activity, representing 76.9% (traditional) and 75.3% (video review), while running and cycling were the least common (~4%–5% of total observed). The ICCs for single measures provided in Table 2 suggest excellent levels of reliability between methods for the total observed, and the numbers of walkers, runners, and cyclists observed, and moderate reliability for sitting/standing.

Figure 4, Figure 5, Figure 6, Figure 7 and Figure 8 show the Bland–Altman plots comparing traditional BWM and expert reviews of BWM videos. The numerical results associated with these plots are shown in Table 3. Examining the plots visually, wide LOA and relatively large biases are seen for the total observed (Figure 4); however, the difference between methods was not linearly related to the average of the two methods (t = 0.03; *p* = 0.98). Observations for walking (Figure 5) are similar to the plot for total observed (Figure 4) in that the LOA are wide and the bias is relatively large. Although most estimates lie within the LOA, there is a tendency for overestimation by the traditional BWM when observations included large groups of people. The regression results support this finding (t = 2.2; *p* = 0.04). The plot for running is displayed in Figure 6, and reveals excellent agreement between methods when few runners are observed and considerable disagreement when large numbers of runners are observed. This suggests that greater divergence occurs between methods when more runners enter the observation period with the traditional BWM underestimating the actual number of runners. The regression results are consistent with this finding (t = −14.8; *p* < 0.001). Figure 7 contains the plot for cycling. Very few cyclists were observed (no more than seven in a 50 min observation period) and both methods provided nearly identical counts at these low numbers with no trend to greater bias observed (regression: t = 0.36; *p* = 0.72). The last plot for sitting/standing is shown in Figure 8. The LOA are wide and most of the bias estimates fall within the LOA at low and high counts of sitters/standers. The linear relationship between the methods was not significant (t = –0.65; *p* = 0.53) and the biases become large when eight or more individuals were counted as either sitting/standing relative to when fewer than four individuals were counted and the biases reflect both over and underestimation by the traditional BWM.

## 4. Discussion

This study employed WVD technology to examine the validity of the traditional BWM for assessing the number of people engaging in different types of PA and sedentary behaviors on sidewalks/streets. By obtaining videos at the same time traditional BWMs were performed, the recorded images provided an objective record that ultimately served as the criterion measure to which BWM counts were compared. Overall, the traditional BWM provided accurate counts of walking, running, cycling, sitting/standing, and total number of people. However, Bland–Altman analysis revealed that the traditional BWM provided less accurate counts as the number of individuals observed increased.

A sparse amount of evidence exists pertaining to the validity of the observation method for assessing PA and the evidence that is available attests to the validity of observation for determining physical activity intensity (e.g., number of individuals engaged in moderate-to-vigorous physical activity) [48,49]. To the best of our knowledge, the current study is the first to examine the validity of an observation technique for assessing counts of people engaging in different types of physical activities in an outdoor setting. Although the BWM has excellent inter-rater reliability and some convergent validity, its concurrent validity has not been tested [30,31,32,33]. We used a WVD to record high-resolution videos that were subjected to reviews by two experts on physical activity behavior and its assessment. Computer software functions (e.g., zooming, rewind) allowed the reviewers to manipulate the videos and extract detailed information about individuals in the videos, thus producing a criterion for examining the concurrent validity of the BWM. Results suggest that total counts of individuals engaged in various types of physical activities on sidewalks/streets are valid only when the number of individuals in an observation area is relatively small. On average, this may be acceptable in low and medium walkability areas (as defined by WalkScore) where few individuals are observed in a given time period. In high walkability areas that typically have large numbers of people, the recommendation is to not utilize the traditional BWM, especially if user volume is expected to be high. Alternatively, the BWM could be used in all areas so long as it is accompanied with a WVD to provide a means for “checking” BWM counts when user volume is high.

The recommendation for total counts of individuals extends to the specific types of physical activities captured during the BWM. This is especially true for walking and to a lesser extent sitting/standing, which, combined, accounted for ~90% of the total observed. In terms of walking, the traditional BWM provided highly reliable counts when the number counted was low. When the number of walkers seen increased above 50 walkers, significant overestimation by the BWM was found. This could have implications, especially when evaluating community-level PA interventions given that walking is the most common form of PA and most walking occurs on sidewalks/streets [21,50]. Sitting/standing behaviors are assessed with the BWM to provide an estimate of the prevalence of sedentary activity on sidewalks/streets. As with walking, the traditional BWM provided less accurate counts when larger numbers of sitters/standers were present during and observation period.

There are two possible reasons for the increased inaccuracy of the traditional BWM with larger numbers of people. First, there is the difficulty encountered by a field observer when tasked with counting large numbers of people who cross the line of observation (meet criteria for being counted) at roughly the same time. In the high walkability areas, the average number of individuals counted during a 50 min observation period was 168 (compared with 6.5 individuals/50 min in lower walkability areas). Video analysis revealed that the field observers simply missed counting people (mostly walkers) during high user volume periods. Second, the field observer must make in-the-moment decisions about counting criteria and, if a large group of people is involved, accuracy can be considerably compromised if they make the wrong decision. This happened in the current study when the field observer did not count a relatively large group of sitters/standers (~22 people) they thought were patrons standing in the outdoor area of a restaurant (the BWM rules state that individuals on private property are not counted). During the review of the video, it was clear that this group was not on the restaurant property, but rather gathered on the sidewalk, necessitating they be counted.

Very few individuals were observed running or cycling, which is consistent with previous studies on the BWM [30,31]. Counts of cyclists between methods were the same, owing to the nature of cyclists (few cyclists performing a very easily recognizable activity). On the other hand, the traditional BWM underestimated runners. Although running/jogging is somewhat distinguishable from other activities (e.g., walking), there is some potential for overlap (runners counted as walkers). However, in the current study, video review revealed that the underestimation of runners during traditional BWMs was due to field observers missing them during high volume observation periods (along high walkability routes). Runners moved quickly through the observer’s field of vision and when the observer was inundated with counting large numbers of people (mainly walkers), runners were missed.

State-of-the-art video technology (as exemplified by the emergence of several WVDs) affords several advantages for studies using direct observation to assess PA within specific settings (e.g., sidewalks/streets). The WVDs are discrete and provide both video and audio data along with date, time, and geo-coordinate information. In addition, videos can be saved indefinitely for subsequent retrospective analysis and they can be examined to detect environmental aspects that may have influenced human utilization of the area observed. Thus, a more robust picture of physical activity can be constructed. Despite these benefits, there are caveats to using video recording behaviors in the field. Reviewing videos is time consuming, but the possibility exists to automate the process using computer vision algorithms, virtually eliminating human involvement in the video review process [34]. Another drawback is the potential for the video recording devices to fail during data acquisition. This occurred a few times in the current study, but the negative effects were minimized by providing data collectors with a backup WVD and training on how to troubleshoot the device. Subject reactivity (Hawthorne effect) is a real possibility when using the observation method. This might be overcome to a certain extent when using the type of WVD we did, given that it is indistinguishable from an ordinary pair of sunglasses. More importantly, the observer can walk normally down the sidewalk/street, whereas the traditional BWM calls for the observer to use a clipboard and record events, which is undoubtedly noticed by passersby. Legal and ethical concerns are often raised when using the observation method in public settings, especially if behaviors are videotaped. However, federal regulations state that research involving observations of human behaviors in public places is acceptable as long as the experimenter does not interact with those being observed and information obtained is recorded in a manner that human subjects cannot be identified (e.g., video images blurred) [51]. It is possible that illicit activities could be captured during video tapping. Our policy is to provide access to our videos, should access be needed by law enforcement during the investigation of a crime and abide by local, state, and federal laws regarding conduct if and when an illegal activity is observed (e.g., failure to report laws).

## 5. Conclusions

In conclusion, the traditional BWM provides valid counts of individuals walking, cycling, running, and sitting/standing on sidewalks/streets in low to high walkability areas. However, when large numbers of individuals are encountered, which is typically the case along high walkability routes, the traditional BWM should not be relied upon to provide accurate data. Video capture and analysis (especially if automated) holds promise for improving the assessment of sidewalk/street use for PA. It has the potential to establish new levels of accuracy, create scalable algorithms that could be used to assess PA in other settings, reduce time/resource requirements (e.g., decrease BWM time), which could ultimately lower costs, and open up the possibility for retrospective analysis. Future research on developing innovative protocols involving video capture and analysis for assessing PA in outdoor spaces, as well as environmental context, are highly warranted.

## Figures and Tables

**Figure 1 ijerph-16-01927-f001:**
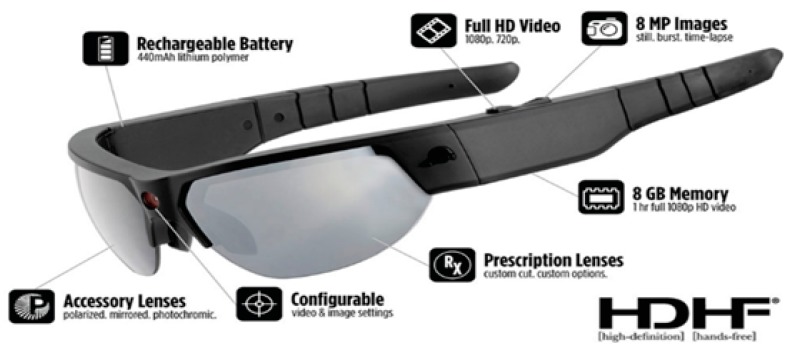
The Pivothead sunglasses used in this study.

**Figure 2 ijerph-16-01927-f002:**
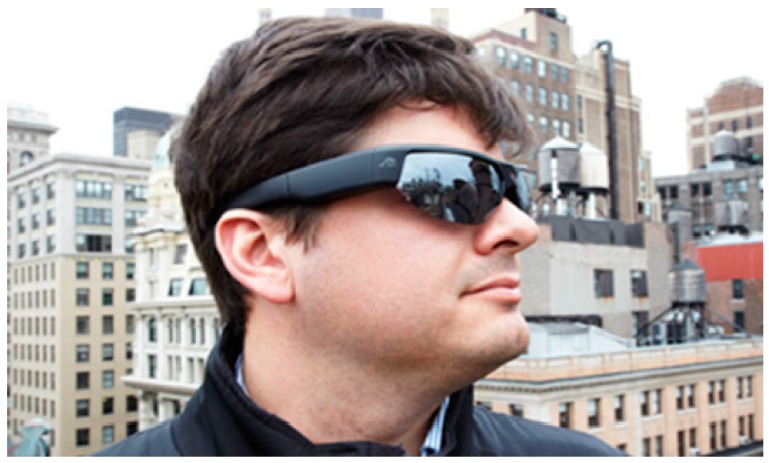
Example of Pivothead sunglasses being worn.

**Figure 3 ijerph-16-01927-f003:**
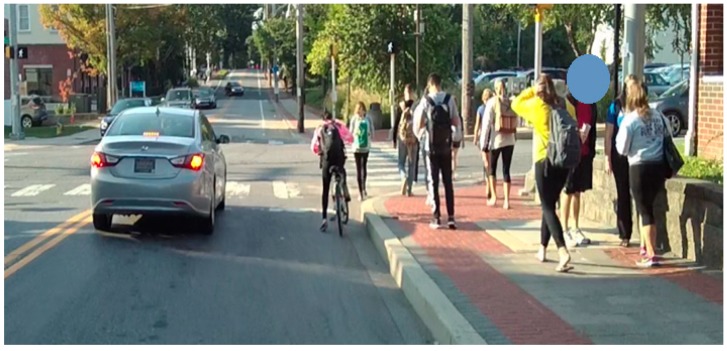
High-resolution image taken with Pivothead glasses.

**Figure 4 ijerph-16-01927-f004:**
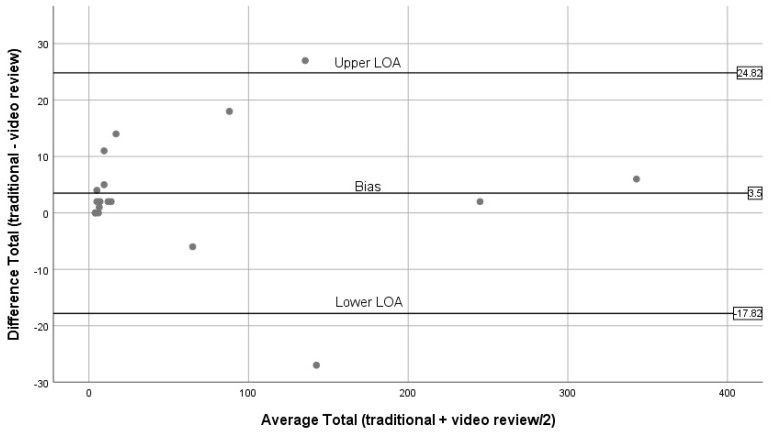
Bland–Altman plots for total (traditional BWM vs. expert review of BWM videos).

**Figure 5 ijerph-16-01927-f005:**
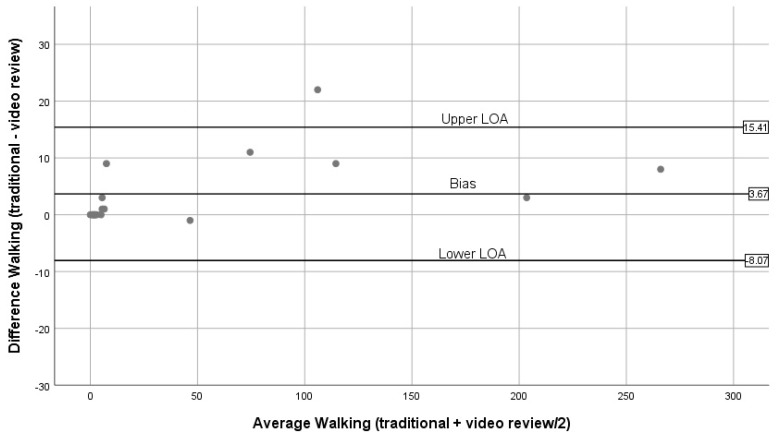
Bland–Altman plots for walking (traditional BWM vs. expert review of BWM videos).

**Figure 6 ijerph-16-01927-f006:**
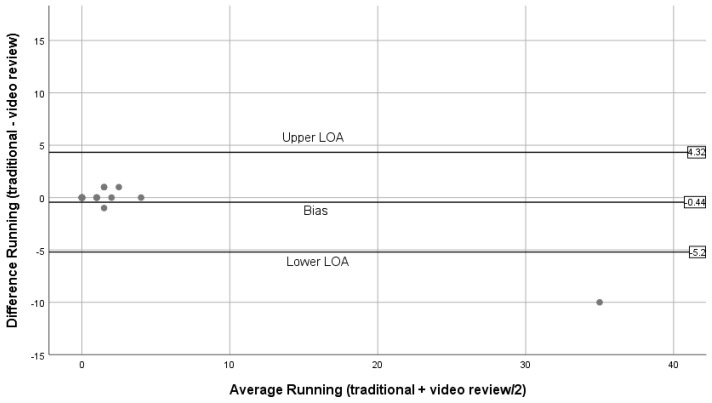
Bland–Altman plots for running (traditional BWM vs. expert review of BWM videos).

**Figure 7 ijerph-16-01927-f007:**
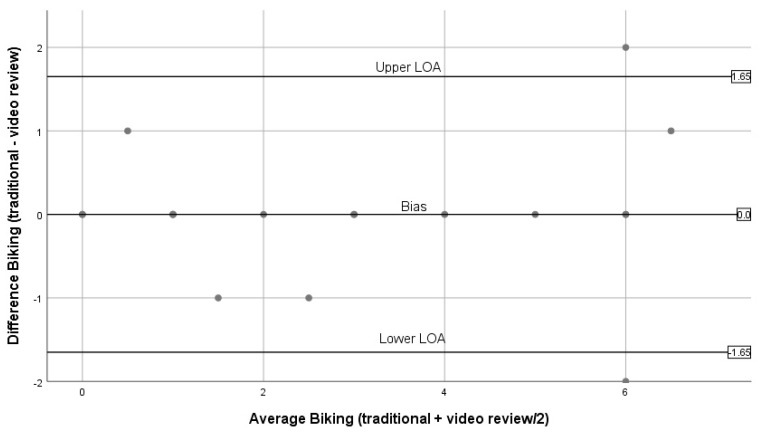
Bland–Altman plots for cycling (traditional BWM vs. expert review of BWM videos).

**Figure 8 ijerph-16-01927-f008:**
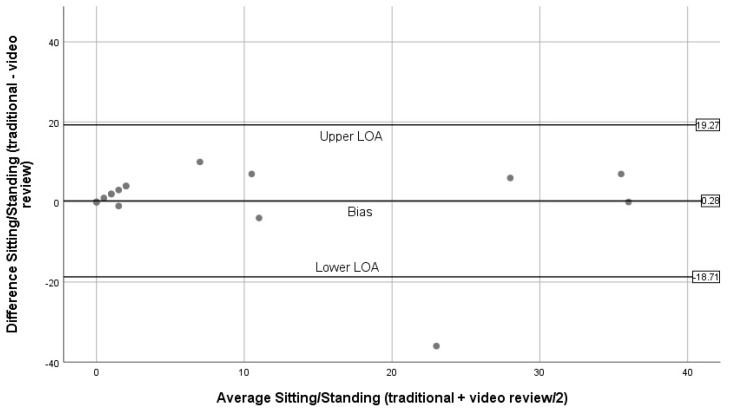
Bland–Altman plots for sitting/standing (traditional BWM vs. expert review of BWM videos).

**Table 1 ijerph-16-01927-t001:** Total numbers of individuals observed during traditional block walk methods (BWMs) and from expert reviews of BWM videos for each BWM outcome.

BWM Outcome	Number Observed (Min–Max/Observation Period)		Percentage of Total Observed (SD)		Mean (SD) ^a^	
	Traditional	Expert Video Review	Traditional	Expert Video Review	Traditional	Expert Video Review
**Walking**	885 (0–270)	819 (0–262)	76.9	75.3	49.2 (79.7)	45.5 (76.9)
**Running**	47 (0–30)	55 (0–40)	4.1	5.1	2.6 (6.9)	3.1 (9.3)
**Biking**	55 (0–7)	55 (0–7)	4.8	5.1	3.1 (2.4)	3.1 (2.3)
**Sitting/Standing**	163 (0–39)	158 (0–41)	14.2	14.5	9.1 (12.8)	8.8 (14.3)
**Total Observed**	1150 (4–346)	1087 (3–340)	-	-	63.9 (97.2)	60.4 (97.1)

BWM—block walk method; SD—standard deviation; ^a^ per 50 min observation period.

**Table 2 ijerph-16-01927-t002:** Comparisons between traditional BWM and expert reviews of BWM videos using intraclass correlation coefficients and paired *t*-tests.

BWM Outcome	ICC Single Measure	95% CI
**Walking**	0.997 **	0.992–0.999
**Running**	0.956 **	0.887–0.983
**Biking**	0.937 **	0.841–0.976
**Sitting/Standing**	0.745 **	0.439–0.896
**Total Observed**	0.994 **	0.983–0.998

** *p* < 0.001. BWM—block walk method; ICC—intraclass correlation coefficient; CI—confidence interval.

**Table 3 ijerph-16-01927-t003:** Numerical results within Bland–Altman plots.

Pairwise Comparisons (Traditional vs. Expert Video Review)	Mean Difference	Standard Deviation	Lower LOA	Upper LOA
**Total Seen**	3.50	10.88	−17.82	24.82
**Walking**	3.67	5.99	−8.07	15.41
**Running**	–0.44	2.43	−5.20	4.32
**Cycling**	0.00	0.84	−1.65	1.65
**Sitting/Standing**	0.28	9.69	−18.71	19.27

LOA—level of agreement; Mean difference ± 1.96 × standard deviation.

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
