# Peer review of "Validation of the Block Walk Method for Assessing Physical Activity occurring on Sidewalks/Streets"

_ijerph, 2019, doi:10.3390/ijerph16111927_

Round 1

Reviewer 1 Report

The well-written and easy to understand manuscript provides interesting information about the concurrent validity of the established BWM for assessing physical activity. The statistical approach is appropriate given the study aims. The conclusions are in line with the results reported. Overall, this is an interesting and strong paper. A few minor edits are needed:

1) page 1, line 20, I suggest that the word "seen" be replaced with the word "observed."

2) page 2, line 58, the word "physical" should be "physically."

3) Table 1 headers need to be formatted to be more readable

Author Response

Reviewer 1

Comment 1: The well-written and easy to understand manuscript provides interesting information about the concurrent validity of the established BWM for assessing physical activity. The statistical approach is appropriate given the study aims. The conclusions are in line with the results reported. Overall, this is an interesting and strong paper. A few minor edits are needed:

Response: Thank you

Comment 2. page 1, line 20, I suggest that the word "seen" be replaced with the word "observed."

Response: Agreed, replaced “seen” with “observed”; Page 1 line 21

Comment 3. page 2, line 58, the word "physical" should be "physically."

Response: changed physical to physically; Page 2 line 59

Comment 4. Table 1 headers need to be formatted to be more readable

Response: Hopefully the header is more readable in the revised version

Reviewer 2 Report

The authors present a well-designed study validating a new wearable video device (WVD) compared to the traditional block walk method (BWM) for assessing physical activity on sidewalks/streets. Assessing physical activity use in the built environment is an important area of public health research. Although the results are not surprising, they provide strong support for WVD, particularly for during times of high volume.

The only concern I had was the use of parametric statistics (t-tests) for count data and proportions.  The authors should consider why they did not use statistical methods more appropriate to the data.  Although unlikely to substantially change the results of the study, it seemed odd to use t-tests in those cases.

Minor concerns:

On line 123 the reference has a type (i.e., 42-42)

For Table 1, the formatting for the column headers needs attention.

Congratulations to the authors on a well done paper.

Author Response

Reviewer 2

Comment 1. The only concern I had was the use of parametric statistics (t-tests) for count data and proportions.  The authors should consider why they did not use statistical methods more appropriate to the data.  Although unlikely to substantially change the results of the study, it seemed odd to use t-tests in those cases.

Response: We agree and have removed the t-test information and now use the intraclass correlation only. Page 5 line 189 (removed paired t-test from statistical approach) and Page 6 lines 130-132 removed t-test results (also from table 1).  In addition, we added the 95% CI for the ICCs to Table 2.

Comment 2. On line 123 the reference has a type (i.e., 42-42)

Response: Page 3 line 124; changed 42-42 to 40-41 which is now correct

Comment 3. For Table 1, the formatting for the column headers needs attention.

Response: I think that the column headers should be fine in the revised manuscript. At first they didn’t show when I looked at them, then they appeared.

Comment 4. Congratulations to the authors on a well done paper.

Response: Thank you

Reviewer 3 Report

This study aimed to validate the BWM and examine the potential for using a wearable video device (WVD) to assess PA occurring on sidewalks/streets. However, traditional BWM provides valid counts of individuals walking, cycling, running, and sitting/standing on sidewalks/streets in low to high walkability areas. Thus, few advantage is demonstrated with video capture. And I suggested to submit this paper for a journal related to urban heath.

-I suggest to do Cronbach´s alpha test

-Please to add sample size calculus

Author Response

Reviewer 3

Comment 1. -I suggest to do Cronbach´s alpha test

Response: We have changed the statistical approach and have eliminated the use of paired t-test as suggested by another reviewer. To compare counts we used intraclass correlation which is most appropriate for our comparisons (see Zaki et al A Systematic Review of Statistical Methods Used to Test for Reliability of Medical Instruments Measuring Continuous Variables IJBMS, 2013; 16:803-807).

Comment 2. Please to add sample size calculus

Response:  We inserted our calculations Page 5 lines 195-197

Statistical power was 0.8 given 18 observation periods, two observations/observation period, alpha = 0.05, acceptable reliability = 0.7, and expected reliability = 0.9.

Round 2

Reviewer 1 Report

The authors made requested revisions. Thank you.

Reviewer 2 Report

I think the authors did a credible job responding to the suggested revisions.  I recommend acceptance

Reviewer 3 Report

ok